# Monitoring Inland Water Quantity Variations: A Comprehensive Analysis of Multi-Source Satellite Observation Technology Applications

Zhengkai Huang [1], Xin Wu [1], Haihong Wang [2], Cheinway Hwang [3] and Xiaoxing He [4],*

[1] School of Transportation Engineering, East China Jiaotong University, Nanchang 330013, China; zhkhuang@whu.edu.cn (Z.H.); 2022138085704007@ectju.edu.cn (X.W.)
[2] School of Geodesy and Geomatics, Wuhan University, Wuhan 430079, China; hhwang@sgg.whu.edu.cn
[3] Department of Civil Engineering, National Yang Ming Chiao Tung University, 1001 Ta Hsueh Road, Hsinchu 300, Taiwan; cheinway@nycu.edu.tw
[4] School of Civil and Surveying & Mapping Engineering, Jiangxi University of Science and Technology, Ganzhou 341000, China
* Correspondence: xxh@jxust.edu.cn

**Abstract:** The advancement of multi-source Earth observation technology has led to a substantial body of literature on inland water monitoring. This has resulted in the emergence of a distinct interdisciplinary field encompassing the application of multi-source Earth observation techniques in inland water monitoring. Despite this growth, few systematic reviews of this field exist. Therefore, in this paper, we offer a comprehensive analysis based on 30,212 publications spanning the years 1990 to 2022, providing valuable insights. We collected and analyzed fundamental information such as publication year, country, affiliation, journal, and author details. Through co-occurrence analysis, we identified country and author partnerships, while co-citation analysis revealed the influence of journals, authors, and documents. We employed keywords to explore the evolution of hydrological phenomena and study areas, using burst analysis to predict trends and frontiers. We discovered exponential growth in this field with a closer integration of hydrological phenomena and Earth observation techniques. The research focus has shifted from large glaciers to encompass large river basins and the Tibetan Plateau. Long-term research attention has been dedicated to optical properties, sea level, and satellite gravity. The adoption of automatic image recognition and processing, enabled by deep learning and artificial intelligence, has opened new interdisciplinary avenues. The results of the study emphasize the significance of long-term, stable, and accurate global observation and monitoring of inland water, particularly in the context of cloud computing and big data.

**Keywords:** inland water; multi-source satellite observation technology; scientometrics; CiteSpace

## 1. Introduction

Inland water refers to water resources including surface water such as ice and snow, rivers, lakes, and groundwater. Changes in inland water reflect the comprehensive impact of natural factors such as regional precipitation, runoff, evapotranspiration, and human activities, as well as important factors affecting the global water cycle [1,2]. In recent years, benefiting from the development of remote sensing technology and improvements in computer and cloud computing capabilities, multi-source satellite Earth observation technology has achieved unprecedented success in inland water monitoring [3]. Extensive regional and global studies have generated valuable insights for understanding water cycle processes and guiding water resource management decisions.

Satellite technology has become widely utilized for monitoring changes in inland water. The Gravity Recovery and Climate Experiment (GRACE) has been applied to studying mass migration and calculating terrestrial water storage [4–7]. Satellite altimetry

technology is used for monitoring global sea levels, lake levels, and glacier elevations [8,9]. Remote sensing imagery is employed for extracting information on surface water [10,11]. Some scholars have analyzed the research results of satellite technology in the field of inland water monitoring by reading a large number of studies. This traditional review method often requires a long time to read the literature, and the accuracy of the analysis is highly dependent on the author's experience. Moreover, these articles often only cover a single research direction, which can only provide a macroscopic qualitative description and reveal certain regularities and conclusions. Therefore, traditional literature reviews make it difficult to quantitatively and systematically reveal the development process, and the conclusions lack objectivity [12,13].

The scientific knowledge graph, a method used in scientometrics and information metrics, is capable of uncovering the origin and development patterns of knowledge. It visually represents the structural relationships and evolution of knowledge in related fields [14]. For instance, Yang et al. used scientometric methods to summarize 50 years of satellite altimetry technology research and quantitatively analyze the relationship between technological progress and research trends, offering a clear explanation of the overall development and future directions of satellite altimetry [15]. Similarly, Xu et al. conducted a bibliometric analysis of 998 relevant studies from the Web of Science core collection, constructing a scientific knowledge graph to reveal the future development trends of the normalized difference vegetation index (NDVI) [16]. The knowledge graph they constructed enhances the intuitive and concrete description, benefiting both researchers with limited experience and readers seeking a quick understanding of a specific field.

Therefore, we employed CiteSpace to quantitatively analyze the literature regarding multi-source satellite Earth observation technology in inland water monitoring. The main work of this paper includes (1) statistics and trends of the number of publications based on the dataset; (2) statistics on the quantity of basic information in publications; (3) highly cooperative countries and author groups; (4) highly co-cited journals, authors, and literature; (5) clustering and burst analysis based on keyword co-occurrence; and (6) knowledge extraction based on feature words. In this paper, we summarize the existing literature, systematically reveal the development trends and the law of change in this field, and provide guidance and references for further research.

## 2. Materials and Methods

### 2.1. Data Collection

The search topic in the Web of Science core collection was set as TS = (RS OR Remote Sensing OR Satellite Altimetry OR Gravity OR GRACE) AND (River OR Fluvial OR Lake OR Glaciers OR Ice OR Snow OR Wetland OR Groundwater OR Swamp OR Marsh OR Estuary OR Bayou). A total of 30,212 documents published between 1960 and 2022 were refined, including articles and review articles. Table 1 presents the basic information of the dataset, which includes 87,035 authors from 8297 different institutions and 1919 journals in 183 countries and regions. Among the 30,212 articles, the total numbers of citations and quotes at the time of data collection were recorded. On average, each article cited around 50 other articles, and each article was cited by an average of 26–27 articles.

**Table 1.** Basic information of the Web of Science (WOS) core collection dataset.

| Type | Value/Number |
| --- | --- |
| Documents | 30,212 |
| Authors | 87,035 |
| Countries/Regions | 183 |
| Institutions | 8297 |
| Sources | 1919 |
| Average Times Citing per Item | 49.62 |
| Average Times Cited per Item | 27.31 |

### 2.2. Scientometric Analysis

Scientometric analysis of literature is a method of literature analysis that uses bibliometric principles to analyze relevant literature. It involves using mathematical and statistical methods to study the distribution structure, quantitative relationships, and change patterns of the literature. Scientometrics is the study of the quantitative aspects of the process of science as a communication system. It is centrally, but not only, concerned with the analysis of citations in the academic literature. In recent years, it has come to play a major role in the measurement and evaluation of research performance [17].

CiteSpace is one of the common software packages used for scientometric analysis. It is a literature analysis package developed by Professor Chen Chaomei based on Java to conduct statistics, analysis, and mining of the literature on a specific subject. It aims to identify evolutionary trends, development frontiers, and research hotspots within a subject area and generate knowledge network maps based on the analysis results [18,19]. This software has been updated by more than 30 versions. For scientometric analysis, VOSviewer is also a software package that is often used by researchers. VOSviewer was developed by Nees Jan van Eck and Ludo Waltman of the Centre for Science and Technology Studies at Leiden University. Each of these two packages has its own characteristics, and there is a lot of literature that discusses the algorithms and results of both packages. CiteSpace features a time-series-based visualization that can be used to detect trends in subject matter over time and to further predict trends in the subject matter. VOSviewer software mainly uses distance-based visualization methods to draw maps by limiting the relative positions between texts and has strong knowledge graph presentation capabilities [20–22]. In general, the two software packages differ only in their different functions. The purpose of both is the visualization of textual data, so for the analysis in this paper, we used CiteSpace (6.1.R4) and VOSviewer (1.6.18), with CiteSpace as the main package and VOSviewer as a supplement.

### 2.3. Processing Flow

We first downloaded and formed a dataset of publications that met the requirements of this study. The basic information of this dataset, including year, country, institution, journal, and author, was first counted quantitatively. Next, countries and authors were analyzed using co-occurrence analysis to obtain the cooperative relationship. Co-citation analysis was then used to illustrate the influence of journals and authors, and the resulting co-citation network of literature can reveal the evolution of research themes. To obtain the clustering graph and explore the frontiers of the field, we used co-occurrence analysis and burst analysis on keywords. We detected word frequency based on feature words (title, keywords, and abstract) and obtained the interannual variation in the number of study themes and areas. The flow schematic is shown in Figure 1 below.

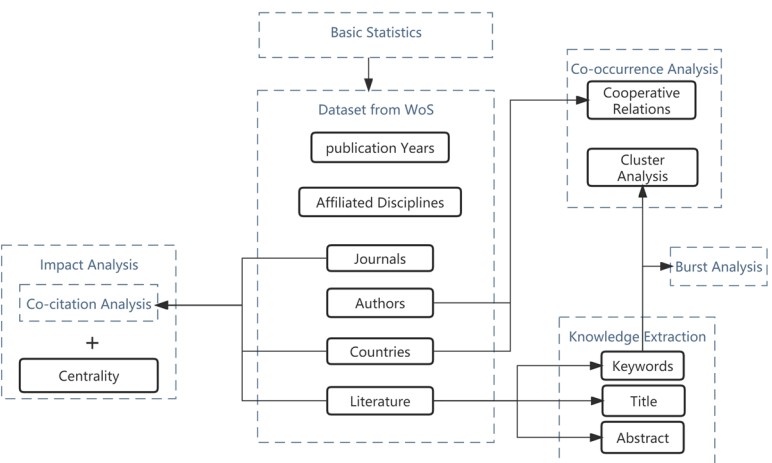

**Figure 1.** Schematic diagram of scientometric analysis.

## 3. Results

### 3.1. Annual Statistics of Publications

We counted 30,212 papers that fit the research theme from 1960 to 2022. In terms of the time series of published papers, it can be roughly divided into three phases, i.e., the start-up, steady growth, and rapid growth phases, as shown in Figure 2. (1) The period 1960–1990 was the start-up stage, when satellite technology was just being developed and was still in the technology verification period. Launched satellites, such as Geosat, Skylab, etc., generally had low accuracy and unstable performance. Therefore, there were no large-scale scientific applications of these satellite data, and thus the number of publications in this period was less than 100. (2) The period 1990–2005 was the steady growth phase. This phase is represented by the T/P family, the ERS family, and Landsat-5. Due to the improvements in satellite orbiting accuracy and sensor accuracy, the stability and accuracy of satellites were greatly improved during this period. As a result, satellite observation technology was gradually used for inland water monitoring, and the number of related studies increased slowly. (3) The third phase, from 2005 to the present, is the rapid growth phase. This phase has involved the development of multiple satellites and accumulated nearly 40 years of continuous observation data. Satellites such as the Interferometric Synthetic Aperture Radar (InSAR) and Global Navigation Satellite System (GNSS) have also been gradually used for inland water monitoring [6]. Therefore, the number of studies has grown rapidly in this phase.

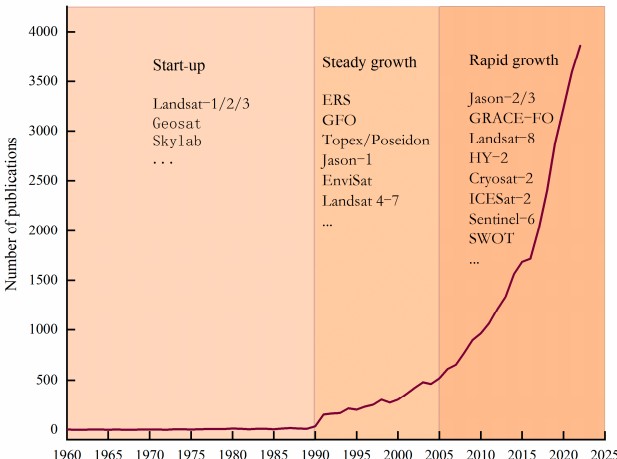

**Figure 2.** Statistics on the number of publications based on the WOS database.

### 3.2. Basic Information Statistics

3.2.1. Basic Statistics on the Number of Countries, Institutions, and Authors

According to the analysis, a total of 183 countries published relevant literature. There are 10 countries that have more than 1000 publications: the United States (9424), China (8960), Germany (2508), Canada (2096), the United Kingdom (2056), India (1977), France (1973), Italy (1386), Australia (1374), and the Netherlands (1082). It is estimated that more than 80,000 authors and 8000 affiliations have made contributions to this field. As shown in Figure 3, the Chinese Academy of Sciences (CAS), National Aeronautics and Space Administration (NASA), Centre National de la Recherche Scientifique (CNRS), and Helmholtz Association contributed 34% of the literature, which represents the main portion of the publications. In addition to these institutions, universities such as the University of California, the University of Colorado, the University of the Chinese Academy of Sciences, and Wuhan University in China have also published more than 600 articles. The top 10 authors by the number of publications are also shown in Figure 3 below, including Shum CK (83), Ma RH (79), Li YM (69), Duan HT (68), Hu CM (63), Bresciani M (60), Pradhan B (58), and Song KS (53).

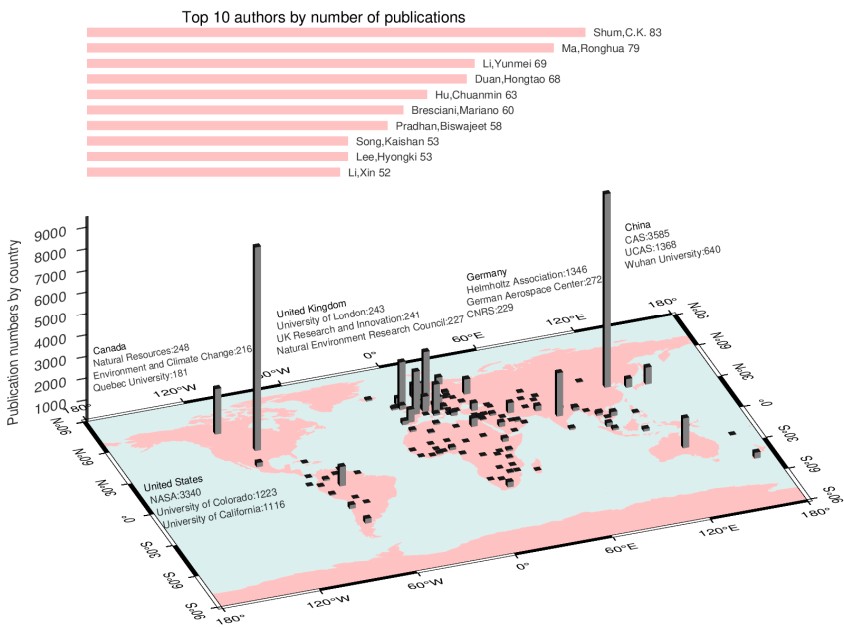

**Figure 3.** Top 10 authors of publications and geographic distribution of the literature collected by category of affiliation.

### 3.2.2. Basic Statistics on the Number of Affiliated Disciplines

These publications are divided into 162 disciplines according to the Web of Science categories. The information provided suggests that out of the 30,212 papers analyzed, the majority (over 70%) are distributed among several disciplines. The highest percentage (23.45%) falls into the category of Earth multidisciplinary science. Other significant disciplines include environmental science (19.72%), remote sensing (11.27%), imaging science and technology (9.31%), and water resources (9.04%). The remaining 30% of publications cover a range of disciplines such as geophysics, marine science, atmospheric meteorology, ecology, geology, astrophysics, civil engineering, etc. Figure 4 shows statistics on the main subject groups to which publications belong in different countries. It appears that the focus of research varies across different countries. Publications in the United States, Germany, Canada, the United Kingdom, and France focus mainly on multidisciplinary geosciences, while those in China and India are mostly in the field of environmental sciences. This illustrates the differences in research priorities and academic traditions between countries. This difference is also due to the fact that China and India are populous countries and face greater geographical and water resource management pressures. As a result, these countries are paying more attention to inland water monitoring and environmental protection, which may lead to more research focusing on environmental science.

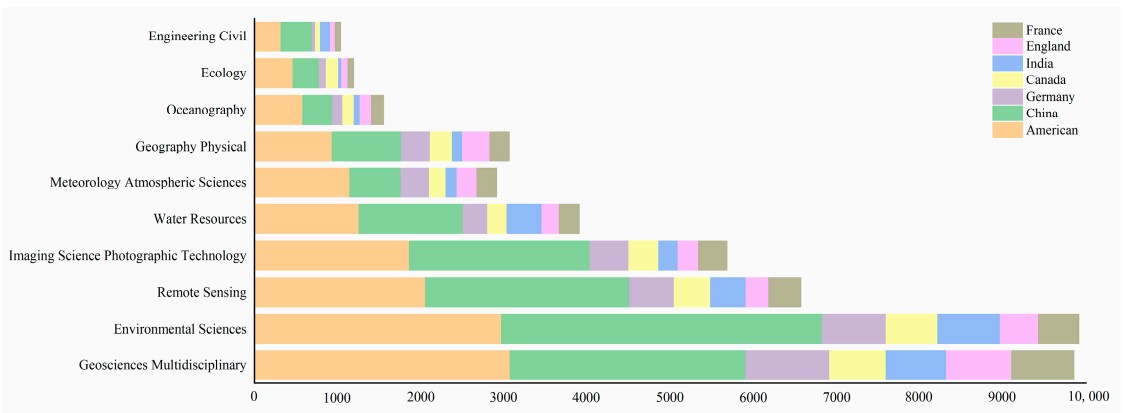

**Figure 4.** Number of publications by subject category by country.

### 3.2.3. Basic Statistics on the Number of Journals

The literature was collected from approximately 1919 different journals. Of these journals, two stand out, each with more than 1000 articles. The two journals are Remote Sensing (2373 papers) and Remote Sensing of Environment (1026 papers). Additionally, several other journals have published more than 600 articles each. These include the International Journal of Remote Sensing (718), *Geophysical Research Letters* (631), *Journal of Hydrology* (619), IEEE Transactions on Geoscience and Remote Sensing (614), and Water (547). Figure 5 presents these statistics. Remote Sensing is one of the leading sources of literature on the application of multi-source satellite Earth observation techniques in terrestrial hydrology, with the largest collection of literature. Remote Sensing of Environment focuses on biophysical quantitative methods in respect of terrestrial, oceanic, and atmospheric transport [23,24], with an impact factor of 13.66, making it the most cited journal. The International Journal of Remote Sensing focuses on remote sensing of the atmosphere, biosphere, cryosphere, and Earth, as well as human modifications to the Earth system [25]. The *Journal of Hydrology* publishes original research papers and comprehensive reviews across all subfields of the hydrological sciences. It includes water-based management and policy issues that affect the economy and society. Science and Nature are known for featuring the latest advancements and development trends in various fields. The articles published in these two journals can provide significant references and spark new ideas for scholars. It is important to note that the information provided is based on the specific journals mentioned in the context of the research analysis. There are numerous other journals in different disciplines that contribute to the overall body of literature.

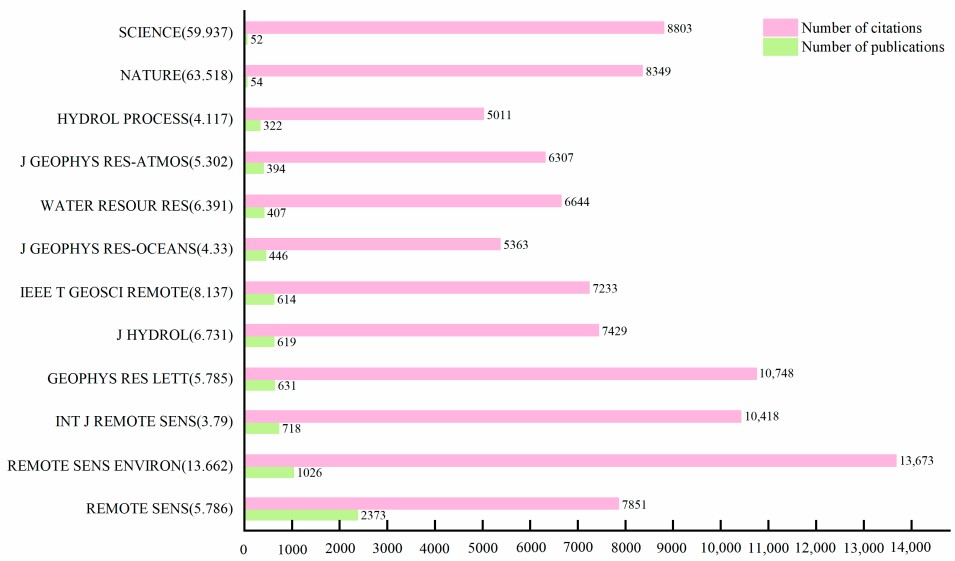

**Figure 5.** The most published journals and their citations.

### 3.3. Cooperative Relations

#### 3.3.1. National Cooperative Network

Figure 6 shows the network of cooperation between countries. It is clear from the chart that China and the United States are the two countries with the largest numbers of publications and the highest intensity of cooperation with other countries. China has collaborated with 50 other countries on 3951 articles, of which the United States was the most important collaborator with 1306 links, followed by Germany, Australia, Canada, the Netherlands, and the United Kingdom. The United States, in addition to its close collaboration with China, has established significant partnerships with countries such as the United Kingdom, Germany, France, and Canada. The United States is the most cooperative country, with a total link strength of up to 6646. Indeed, there is a clustering of Denmark, Iceland, Luxembourg, Sweden, and New Zealand into a group, as well as

the grouping of Germany, Poland, the Czech Republic, Switzerland, Finland, Hungary, Spain, Italy, and other countries. This phenomenon indicates the influence of regions on the similarity of countries in terms of research collaboration. Geographically close countries in Europe tend to exhibit higher similarities in research collaboration.

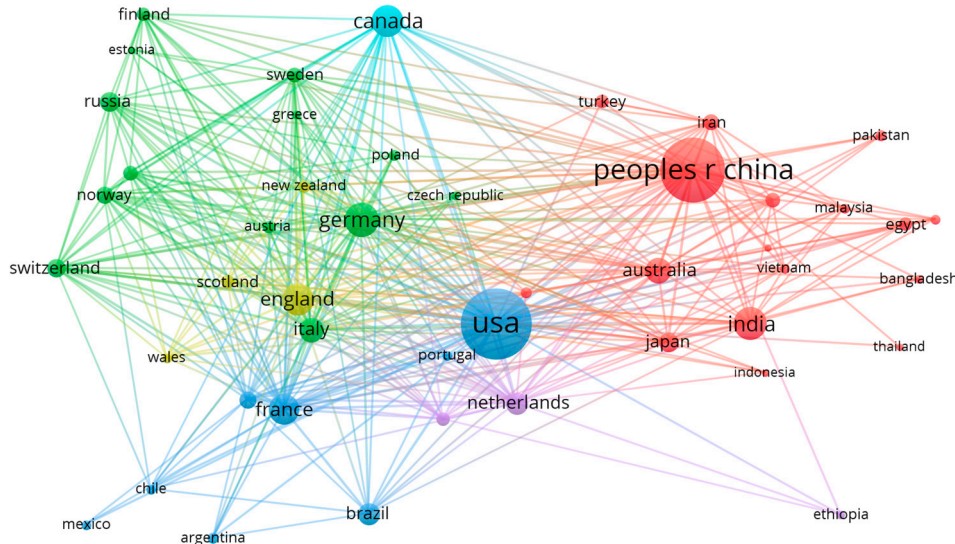

**Figure 6.** National cooperation networks. A network of 46 countries with more than 100 collaborative papers. The nodes are divided into five groups, each represented by a different color.

### 3.3.2. Author Cooperative Network

The authors with the highest numbers of collaborative articles are Shum CK (66), Ma RH (63), Hu CM (48), Li YM (47), Bresciani M (44), and Duan HT (43). Figure 7 reflects that there are two core author groups. The core author group formed by Shum CK and Lee H mainly studies the application of satellite altimetry technology and satellite gravity, particularly utilizing GRACE data for regional surface water monitoring and establishing hydrodynamic models [23–26]. The second group consists of Duan HT, Hu CM, Song KS, Li YM, and Ma RH [27–30] and mainly uses visible light remote sensing images to detect inland lakes and interpret corresponding hydrological phenomena. Additionally, authors such as Li X, Chen X, and Wang L act as bridges between these two core author groups, promoting the frequency and close connection between authors.

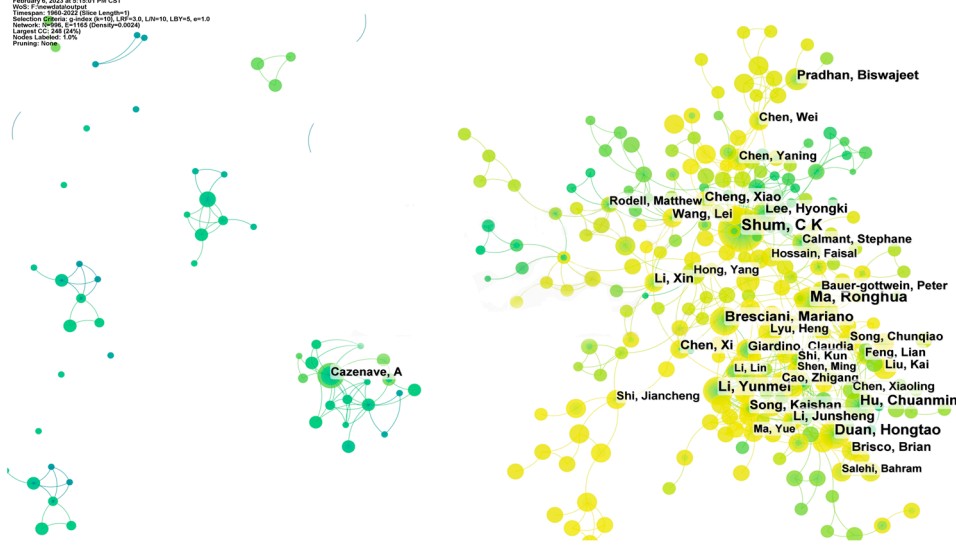

**Figure 7.** Author collaboration network. This network consists of 998 nodes and 1994 links.

Figure 7 also highlights the presence of smaller author groups. One such small author group consists of Cazenave A, Calmant S, Frappart F, Kouraev AV, and Ramillien G [31,32]. Their research is centered around remote sensing and satellite gravity technology applied to river flow and basin-level calculations. Another group includes authors such as van den Broeke MR, Bamber JL, Rignot E, and Berthier E [32–34]. This group focuses on different monitoring methods to monitor and calculate parameters such as the thickness of polar ice sheets, sea ice, and the mass balance of glaciers. These small author groups tend to have more specialized research interests and themes, despite being smaller in size and less popular than the core author groups. They have created new ideas, innovative methods, and unique perspectives in a specific field. Their small size and specific research direction can make them more flexible in collaborative decision-making.

### *3.4. Analysis of the Impact*

### 3.4.1. Co-Citation Analysis of Authors and Journals

Centrality is an indicator that compares the activity of an institution or author with that of other institutions, authors, etc. and acts as a bridge between them. Nodes with a centrality of more than 0.1 in CiteSpace are called critical nodes. Figure 8a shows the top 10 most cited journals and their centrality. Remote Sensing of Environment, Geophysical Research Letters, and the International Journal of Remote Sensing are the three journals with the most citations. Different journals have different focuses, and the literature they include is also different. Science of the Total Environment is an international multidisciplinary journal with the widest range of disciplines. Its centrality exceeds 0.1, which is the highest centrality of all the journals.

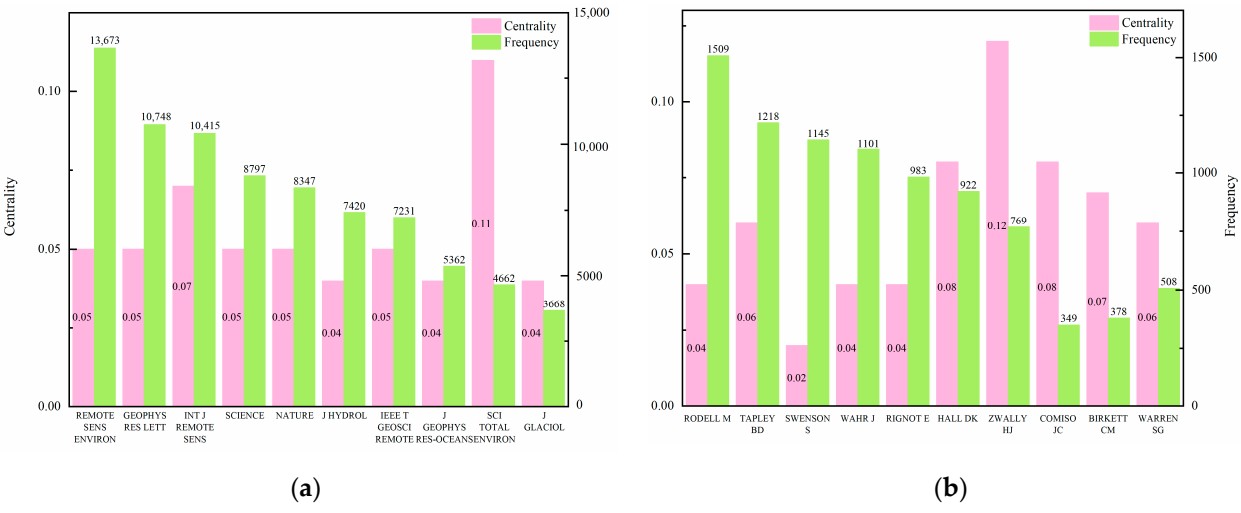

**Figure 8.** Most influential journals and authors: (**a**) the most frequently cited journals and their centrality; and (**b**) the most frequently cited authors and their centrality.

Figure 8b presents the top 10 authors with the highest citation numbers and centralities in the co-citation author network. Rodell M. primarily focuses on studying large-scale or global temporal and spatial changes in the water cycle. This includes analyzing data from GRACE and GRACE-FO to understand the resulting impacts on climate, drought, and water storage [35–37]. With over 1500 citations, Rodell is considered the most influential author in this research field. Tapley BD and Swenson S are another two prominent authors in the field, with 1218 and 1145 citations, respectively. Zwally HJ is the only author with a centrality of more than 0.1 and plays a role in bridging communication between different authors.

### 3.4.2. Literature Co-Citation Network

The co-citation network of the literature takes on a static form with a time series. As shown in Figure 9, in this co-citation network, the article written by Gorelick N. et al. on promoting the Google Earth Engine (GEE) in 2017 is the most cited paper [38]. This means that more users can access and take advantage of the data, tools, and functions provided by the platform. This will make data analysis and processing faster and interdisciplinary and cross-border cooperation stronger. GEE also highlights the use of time-series satellite data for large-scale monitoring of land and water dynamics [2,39]. Pekel JF et al. collected millions of Landsat images on GEE and quantified the global changes in surface water between 1984 and 2015, leading to the production of a comprehensive global map of surface water [40]. In an article cited 386 times in the field, the study identified prolonged drought and human activities as significant contributors to global surface water transformation. Moreover, three of the other top five co-cited papers are about GRACE, including the JPL RL05 mascon and CRS RL05 mascon solutions [41,42]. The gravity field model solved using GRACE data has become an important means for large-scale land mass migration changes, such as global sea level transformation, regional surface water monitoring, polar ice sheet and mountain glacier melting, and seismic coseismic change [5,43,44]. At present, the latest GRACE data have been updated to version RL06. The literature in respect of Ice, Cloud, and land Elevation Satellite-2 (ICEsat-2) and Surface Water Ocean Topography (SWOT) has also been co-cited more than 100 times [45,46]. These technologies, along with other relevant datasets such as ECMWF Reanalysis (ERA5), Bedmap, Modern-Era Retrospective Analysis for Research and Applications (MERRA), Global Land Evaporation Amsterdam Model (GLEAM), and Randolf Glacier Inventory (RGI), contribute to the key knowledge of multi-source satellite Earth observation technology in inland water exploration.

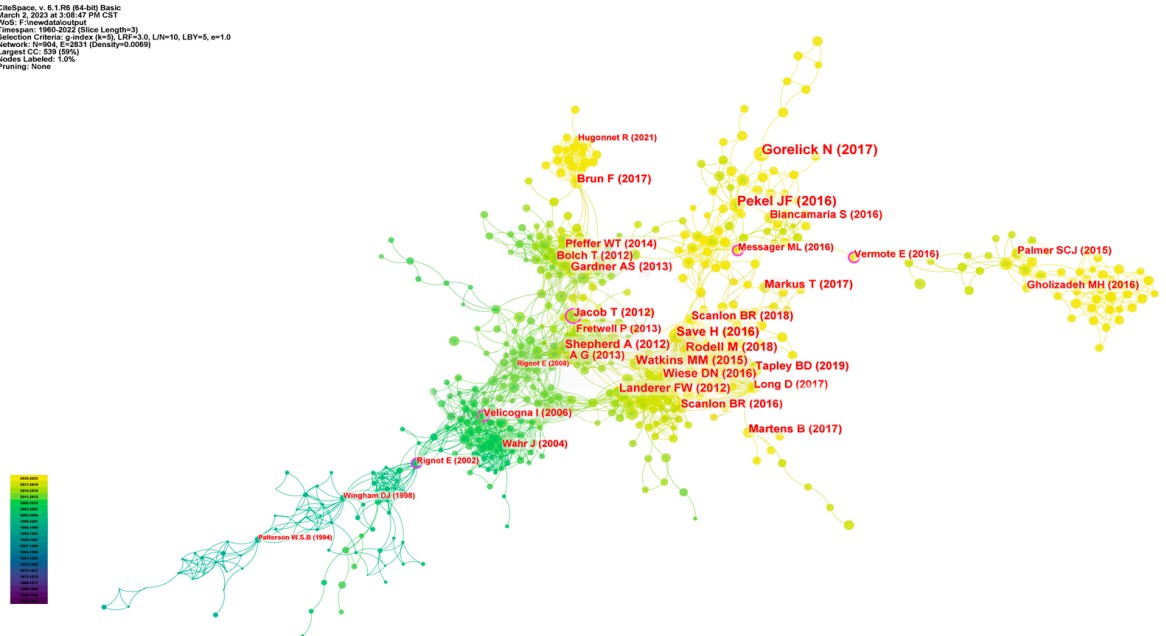

**Figure 9.** Literature co-citation network. The time slice was set to 3 years, and the details display the 33 nodes with the most citations. The literature co-citation network is composed of 904 nodes and 2831 co-citation links.

The co-citation network of literature can not only reflect the citation status but also grasp the development order of the field through basic information such as titles, keywords, and abstracts. From left to right, the theme of evolution in chronological order is represented. As shown in Figure 10, similar literature, such as topics, technologies, or data usage, tends to cluster closely in the network. In the alternation of the observation technology and observation theme, we can see the promotion effect of the advancement of

multi-source satellite Earth observation technology on the monitoring of different components of inland water. Initially, satellite altimetry techniques were predominantly employed for glacier monitoring and global mean sea level estimation, where low accuracy requirements were acceptable. Gravity satellites have significantly enhanced the accuracy of global gravity fields while filling the technical gaps in groundwater monitoring. Long-term monitoring of groundwater in California and the North China Plain has been successfully conducted [47–49]. In recent years, researchers have shifted their attention to the extraction and monitoring of surface water, such as rivers and lakes. Mountain glaciers and lakes on the Qinghai–Tibet Plateau became research hotspots during this period. The launch of SWOT will assist researchers in comprehending and tracking the distribution and changes in water worldwide. This mission will be the first comprehensive global survey of Earth's surface water [40]. In addition, with the support of interdisciplinary techniques based on deep convolution, surface water extraction on a global scale has entered a new phase.

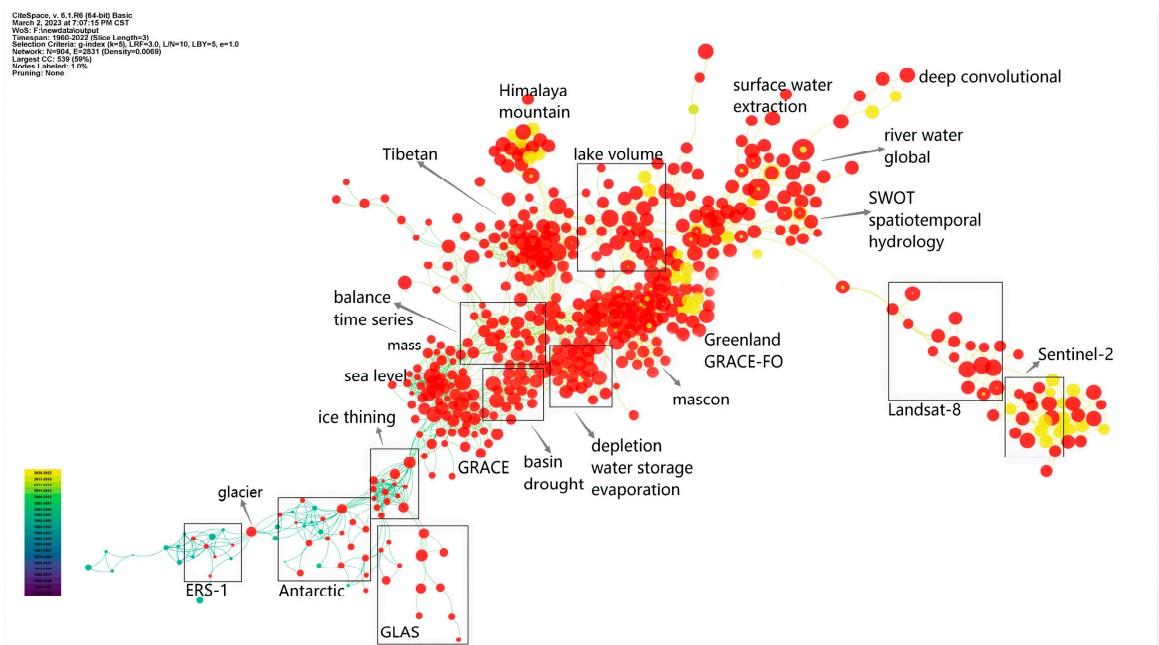

**Figure 10.** Evolution of research themes over time. Literature with high bursts is shown in red.

### 3.5. Keyword Co-Occurrence Analysis

Co-occurrence analysis is a commonly used method in text mining and topic modeling. The principle of co-occurrence analysis is to extract keywords and analyze the connections between them based on their co-occurrence frequency. Figure 11 shows the breakdown of the clusters. Cluster 1 is related to remote sensing image and surface water monitoring and consists of the keywords remote sensing, Landsat, lake, river, dynamic, wetland, basin, classification, geographic information system (GIS), etc. Cluster 2 mainly includes keywords such as model, groundwater, climate change, drought, grace, depletion, prediction, etc., which are related to gravity satellite and groundwater monitoring. Cluster 3 relates to MODIS and monitoring the thickness of snow cover. Cluster 4 is associated with the use of satellite altimetry, including radar and laser measurements, for monitoring ice sheets. It involves ice characteristics, mass balance, and hydrological processes, particularly in regions like Greenland. Cluster 5 is significantly far away from the other four clusters; it appears to be relatively independent and less directly related to the other four clusters. This cluster focuses on the application of InSAR and GNSS for observing land subsidence and its dynamic response to changes in inland water storage.

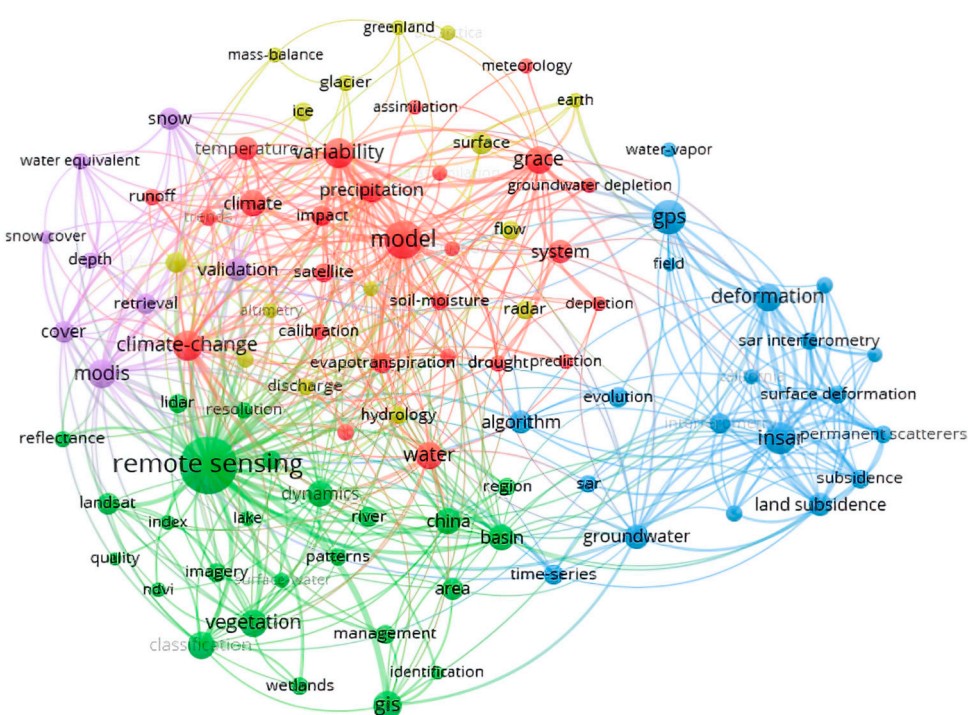

**Figure 11.** Keyword clustering network. The network only shows keywords that have co-occurred more than 120 times. The network contains 94 nodes and 3839 links. These nodes are clearly divided into five clusters, which are represented by different colors.

### 3.6. Knowledge Extraction Based on Feature Words (Titles, Keywords, and Abstracts)

To gain insights into the evolution of research topics and identify frontier themes, we also analyzed the frequency of hydrological phenomena and study areas since the 1990s. The search scope encompasses titles, keywords, and abstracts. The analysis tracks the temporal evolution of these topics.

#### 3.6.1. Research Theme

Figure 12 provides insights into the frequency and trends of hydrological phenomena that are of greatest concern to researchers in the field. Overall, the integration of hydrological phenomena and Earth observation is developing exponentially. Studies on precipitation and temperature have accumulated the largest amount of literature, with precipitation appearing 6474 times and temperature appearing 5097 times. Precipitation-related studies have shown an exponential upward trend over the past 30 years. Precipitation and temperature data are often used as meteorological indicators to assist in various hydrological interpretations, such as evapotranspiration calculations, groundwater storage assessments, surface water flow analysis, and drought studies [50–53]. Although a number of studies on snow and ice melt were recorded early on, their growth has been relatively slow, with a total of 2288 records by 2023. This suggests a sustained but modest interest in this phenomenon. Research on surface water exchange started later, with a total of 1296 records. However, the word frequency between 2018 and 2022 reached 732, surpassing 50% of the total from the previous 28 years. This increase is attributed to the emergence of satellite gravity technology, particularly GRACE-FO. Satellite gravity technology provides a means by which to detect groundwater conditions on a large scale, leading to increased interest and studies in this field [54,55]. The phenomenon of runoff and seasonal change returned approximately 2500 records, showing an overall upward trend from 1990 to 2021, with a slight decrease in the period 2021–2022. Evapotranspiration had a relatively lower frequency, with fewer than 1000 occurrences, and showed a modest increase over the past 30 years. Therefore, it appears to be the phenomenon of least concern among the hydrological phenomena studied. There are many challenges to accurately quantifying evapotranspiration. These include the

time extension of remote sensing data, the scale transformation of remote sensing information, uncertainty in remote sensing classification, and the universality of remote sensing evapotranspiration models and inversion algorithms [56,57]. In addition, the evaporation paradox is proposed, so accurate quantitative monitoring of evapotranspiration will take a long time to verify.

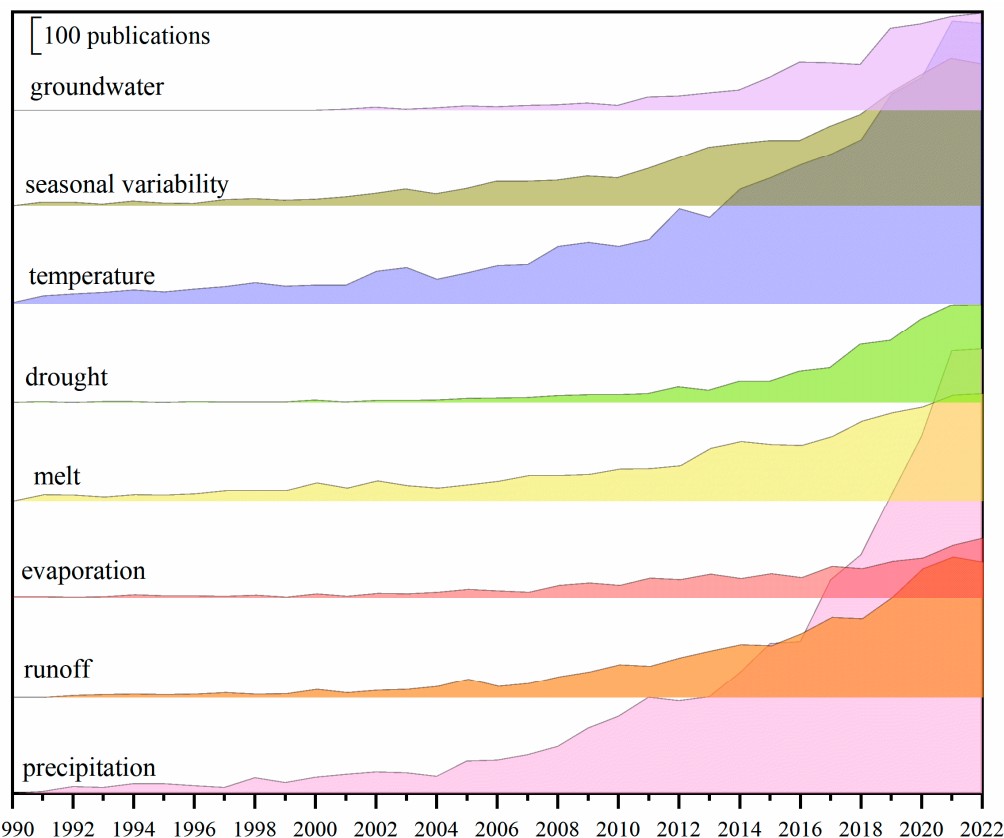

**Figure 12.** Number of hydrological phenomena researched by year.

### 3.6.2. Research Area

Figure 13 depicts the 10 study areas with the highest number of occurrences. They are Antarctica (2153), Tibet (2055), the Yangtze River Basin (1799), the Arctic (1342), Greenland (1273), the Alps (1115), the Indian Basin (956), the Amazon Basin (740), the Mississippi Basin (703), and Alaska (682). Compared with other study areas, the study of continental glaciers such as Antarctica and Greenland started early, especially the exploration of Antarctica, accumulating more than 160 publications before the 21st century. According to statistics, the study of polar glaciers has the most overlapping trends, and both locations have seen a decline in numbers in recent years. In the 21st century, there has been a notable increase in quantitative studies focused on regions such as the Indian Basin, Yangtze River Basin, and Qinghai–Tibet Plateau. These areas are heavily influenced by humans and have a wide-ranging impact, making them popular study areas in recent years. These areas are expected to remain hot research topics in the future. Due to the special geographical location and harsh natural environment, the number of hydrological and meteorological stations on the Qinghai–Tibet Plateau is limited. For a long time, research on the climate response mechanism of lake changes on the plateau was mostly restricted to the qualitative description of precipitation, evaporation, temperature, wind speed, cryosphere melting, and other climatic factors. In the past decades, advancements in technology, recognition of the importance of related research, data-sharing initiatives, and interdisciplinary cooperation have resulted in significant progress in the quantitative study of the Qinghai–Tibet Plateau, making it the most popular study area with the fastest growth rate and most

increments [58]. In the comparison of the time dimension, the results of this statistic show that the main research area in this field has changed from polar glaciers to the Qinghai–Tibet Plateau and large river basins, such as the Yangtze River Basin and the Indian Basin.

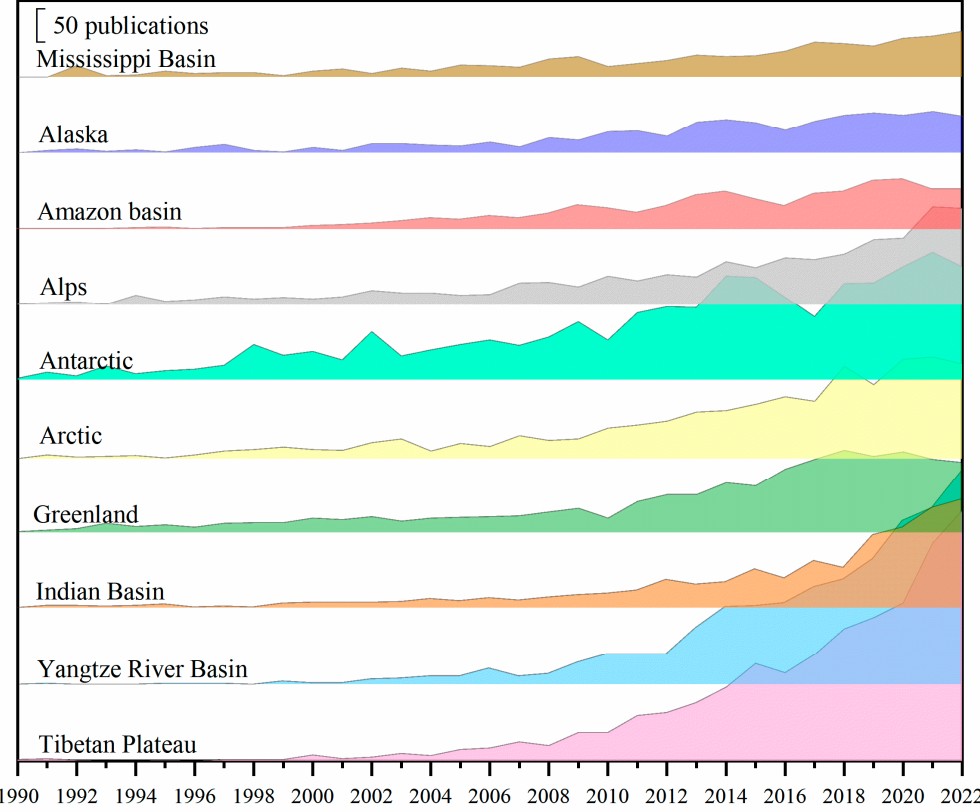

**Figure 13.** Number of different study areas researched by year.

### 3.7. Burst Analysis of Keywords

As shown in Figure 14, the results reveal three distinct phases in the evolution of research topics in this field. During the first phase (1990–2000), a significant number of studies utilized technologies such as satellite altimetry for Earth observation. The acquired information was predominantly used for the development of numerical models and general circulation models. The popular study areas during this phase were primarily focused on large glaciers, including Antarctica and Alaska. This phenomenon echoes the conclusions above. In the second phase (2000–2018), researchers increasingly emphasized the application of related technologies to study surface water, glacier mass balance, hydrology, land cover, and other related aspects. This period involved a surge in publications showcasing the practical applications of remote sensing data. In the third phase (2018–present), improvements in the resolution and accuracy of various sensors and a more complete Earth observation series have accumulated massive amounts of data for scientific research. New technologies, such as automatic identification and intelligent batch processing, have emerged. The latest methods for data processing, including random forests, machine learning, and deep learning, have gained prominence. Researchers have turned their attention to a combination of hydrological phenomena and interdisciplinary data-processing algorithms. Intelligent image recognition and data processing based on the above algorithms have become the forefront of development in this field. Throughout the entire timeline, scholars have demonstrated a longstanding interest in studying the optical properties of various sensors, gravity detection, and sea levels. As shown in Figure 14, the timeline of these three keywords is grayed out. This phenomenon indicates their sustained importance in the field of hydrological research.

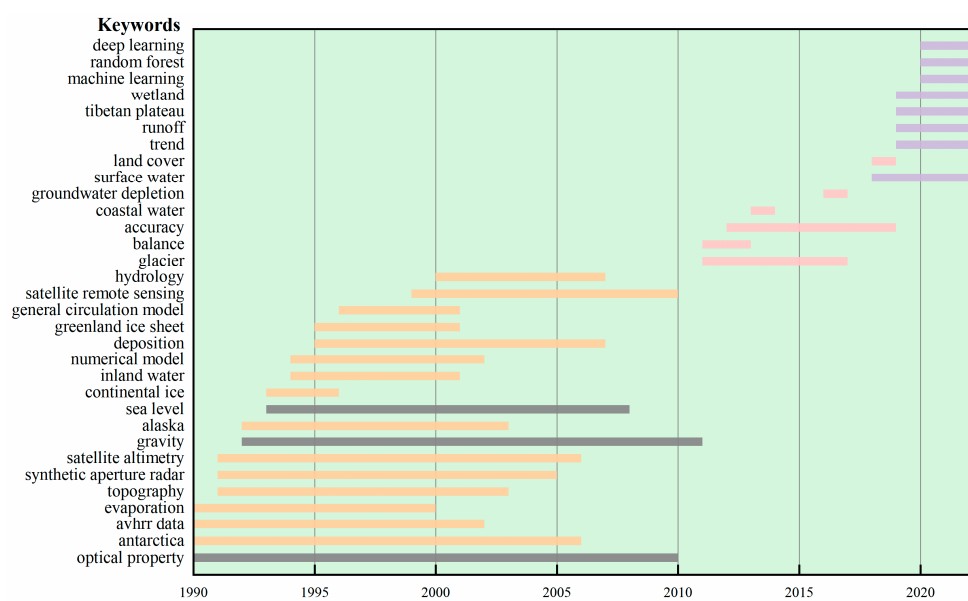

**Figure 14.** Results of keyword burst analysis. The time slice was set at 3 years. The 32 keywords shown in the graph meet the following requirements: they are in the top 8% of each slice and have the highest bursts.

## 4. Discussion

### 4.1. The Role of Time-Varying Quantitative Analysis

One of the contributions of our analysis is the literature on the application of scientometric analysis to the application of multi-source satellite Earth observation techniques to inland waters. This paper is different from the others in that we have added a time-varying quantitative analysis of the research theme and study area, which provides double verification of the evolution of the field and the burst analysis results. This approach enables a more accurate and nuanced quantitative analysis of the evolution and frontiers of the field. Based on the results presented in Sections 3.4.2, 3.6.2 and 3.7, we can determine that the focus in this area has shifted from sea level and polar glaciers to groundwater and surface water. The specific number of changes in relation to different study areas and research topics was quantified. The Tibetan Plateau has been a popular research area in recent years and will continue to be popular in the coming years. Affected by the uneven spatial distribution of freshwater resources in the inflow or outflow basins of the Qinghai–Tibet Plateau caused by climate change, the Yangtze River Basin and the Indian River Basin have become unstable. As the Qinghai–Tibet Plateau is the birthplace of many river basins, the changes in its water environment also have a profound impact on these river basins. Although many scholars have carried out a significant amount of research on climate change and water cycle mechanisms on the Qinghai–Tibet Plateau, there is generally no advanced multi-level coupling model and a lack of ground monitoring [59–61]. Fortunately, the launch of SWOT (https://swot.jpl.nasa.gov/ accessed on 10 July 2023) will effectively improve the monitoring of lake water levels in the region, which will greatly encourage regional and global short-term dynamics and long-term trend monitoring [62,63]. The attention paid to this field will continue to rise in the future.

### 4.2. Validation of the Accuracy of the Results

In contrast to most reviews with a single research theme, the topics covered in this paper belong to a cross-disciplinary field. In interdisciplinary bibliometric analysis, the primary condition for experimental success is to obtain accurate and complete datasets. In order to obtain the literature that best met the requirements of the research topic, we tested how to identify terms that met the criteria for this paper. From broad search terms to precise feature terms, 100 articles in each dataset were selected, and the number of documents and

their accuracy for each search result were determined. The results are shown in Table 2. It can be seen that under more accurate word searches, the number of documents returned is larger and more accurate.

**Table 2.** Different search terms and the number and accuracy of their returns.

| Subject Term of the Retrieval | Number | Accuracy |
|---|---|---|
| TS = Satellite AND Inland Water | 1262 | 70% |
| TS = Satellite AND (River OR Fluvial OR Lake OR Glaciers OR Ice OR Snow OR Wetland OR Groundwater OR Swamp OR Marsh OR Estuary OR Bayou) | 29,292 | 85% |
| TS = (RS OR Remote Sensing OR Satellite Altimetry OR Gravity OR GRACE) AND Inland Water | 1567 | 82% |
| TS = (RS OR Remote Sensing OR Satellite Altimetry OR Gravity OR GRACE) AND (River OR Fluvial OR Lake OR Glaciers OR Ice OR Snow OR Wetland OR Groundwater OR Swamp OR Marsh OR Estuary OR Bayou) | 30,212 | 93% |

In addition, we used Google Scholar to verify the accuracy of citations in the database (Web of Science) used in this paper. The scope of the Web of Science core collection is limited, mainly to SCI, SSCI, ESCI, and other source journals/conference papers with high authority. Google Scholar includes not only the Web of Science but also EI searches, preprint websites (such as ArXiv, SSRN, etc.), and papers such as university/scholar personal websites. Web of Science and Google Scholar calculate the citations of a paper based on how many papers in their databases cite it. We obtained a total of 100 articles with citations of different orders of magnitude from Google Scholar and compared the data with the Web of Science used in this paper. Figure 15 shows the fitted curve, correlation coefficient, and goodness-of-fit for these 100 samples. The goodness-of-fit degree is greater than 0.9, indicating that the straight line fits the sample values well. The correlation coefficient is 0.9775, and there is a strong correlation between the citations in the two databases. This verifies the reliability of the results of this study to a certain extent.

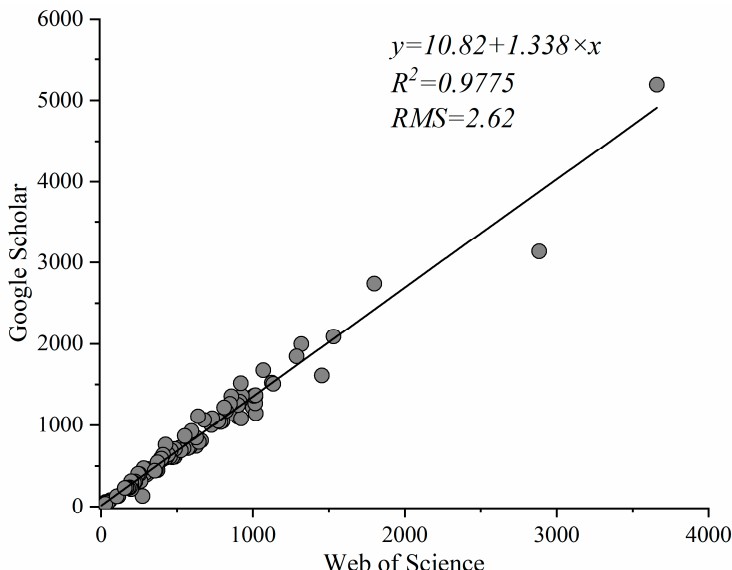

**Figure 15.** Comparison and correlation of citations in Web of Science and Google Scholar.

### 4.3. Detailed Setting of the Software

One potential limitation of this study is that different analysis software and different settings will lead to differences in results. However, as far as the current comparative studies are concerned, the two software packages have been verified many times by different researchers, and the results are considered credible. Pan et al., in their comparison of bibliometric software analysis, showed that most researchers using bibliometric software do not

provide sufficient usage information, which is not conducive to the reader's reproduction of the processing flow and results [20]. Therefore, in this article, under the condition of ensuring accuracy, we have displayed graphical information based on the principle of clear graphical display and as many nodes as possible. The network set by CiteSpace was sliced over a period of 3 years, and the first 10 nodes of each slice were selected to prepare for display. We adjusted the g-index value to actually control the number of nodes to be displayed; the author cooperation network shows a total of 996 nodes, and the document co-citation network displays a total of 904 nodes. Other relevant settings are reflected in the figure name. A small difference in results is allowed, and differences introduced by different settings do not mean errors. We should also take these differences into account when drawing conclusions.

## 5. Conclusions

The analysis of publication trends in this field reveals several key findings. The number of publications in this field has exhibited exponential growth, indicating increasing interest and involvement from researchers. Among the top five publications, the United States, China, Germany, Canada, and the United Kingdom produced more than 80% of the publications. The disciplines and degree of cooperation between countries are also geographically linked. In densely populated countries such as China and India, inland water monitoring mainly serves environmental science, while publications in the United States, Britain, France, Germany, and Canada mainly belong to geodisciplinary fields. In terms of cooperation, China and the United States are the two countries with the highest degree of cooperation, but some European countries, such as Germany, Poland, the Czech Republic, Switzerland, Finland, Hungary, Spain, Italy, etc., also have a strong degree of cooperation. The degree of cooperation is related to geographical distance. In addition, a core group of authors, led by Shum CK and Lee H, has emerged as influential contributors in this field. The development of research in this field can be divided into three periods. The period from 1990 to 2000 marked the exploration phase of new detection technologies. The period from 2000 to 2018 involved significant growth in detection technology development. The period after 2018 represents the development phase of cloud computing and intelligent processing applications. The focus of research has shifted from large-scale glaciers to river basins and lakes.

From the review above, key findings emerge: (1) In recent years, there has been a consistent rise in the number of studies focusing on the Tibetan Plateau and its associated basins, including the Indian Basin and the Yangtze River Basin. The maturity of regional-scale research will inevitably give impetus to the short-term and long-term monitoring of inland water on a global scale. After accumulating a large number of regional inland water studies, inland water studies on a global scale will provide a clearer explanation of the inland water cycle and its mechanisms. (2) Inland water monitoring has grown with advancements in multi-source Earth observation technology, driving research in the field. In particular, the launch of SWOT in December 2022 is expected to be a groundbreaking development in global surface water monitoring. SWOT will provide the first comprehensive survey of the Earth's surface water, offering valuable information for understanding and tracking water resources globally. This mission will generate new momentum. (3) Furthermore, the integration of the hydrological field with disciplines such as deep learning and intelligent recognition is opening up new avenues for researchers. The integration of hydrology with various cross-disciplinary fields is expected to facilitate further exploration and research. Researchers can anticipate a growing number of studies that combine knowledge and expertise from hydrology, cloud computing, big data, deep learning, and intelligent recognition. Through the application of these technologies, outcomes will address the challenges and opportunities associated with the long-term and global-scale observation of inland water. Regrettably, our findings do not capture the relationship between study themes, study areas, and the evolution of research methods due to the absence of quantitative statistics on research methods. To address this limitation, future

scientific analyses on this topic should aim to quantitatively examine research methods and explore the underlying connections between research themes, regions, and methods. Moreover, existing literature analysis software, such as CiteSpace and VOSviewer, lacks advanced keyword refinement and algorithm integration functions, making it challenging for users to obtain accurate results. However, by ensuring a large and comprehensive sample, the influence of individual incorrect samples can be mitigated. Moving forward, we aspire to further enhance natural language processing by integrating multi-source data and strengthening knowledge graph reasoning capabilities.

**Funding:** This work was sponsored by the National Natural Science Foundation of China (41974016, 42104023, 42264001), the Major Discipline Academic and Technical Leaders Training Program of Jiangxi Province (20225BCJ23014), Hebei Water Conservancy Research Plan (2022-28).

**Conflicts of Interest:** The authors declare no conflict of interest.

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
