# Peer review of "Monitoring Inland Water Quantity Variations: A Comprehensive Analysis of Multi-Source Satellite Observation Technology Applications"

_remotesensing, doi:10.3390/rs15163945_

Round 1
Reviewer 1 Report
The authors apply scientometric tools to uncover knowledge's origin and development patterns regarding the application of multi-source satellite technology to monitoring inland waters.
The paper is well written and brings relevant information with a focus on hydrology. I have made a few comments in the attached file.

Author Response
Point 1: The paper is well written and could be published as it is, with minor typing and language review. However, I recommend the authors remove the sentence in lines 66-68 "Compared to traditional review articles, these articles feature more quantitative analysis, comprehensive and systematic research directions, and clear explanations of the overall development trends and frontiers in the field. There is space for both approaches in science, which have their strengths and weaknesses.
Response 1: Thanks to the expert's comments, we have deleted lines 66-68 pointed out by the expert.
Point 2: For instance, the paper title is misleading. When I read the title, I thought that "monitoring inland waters" would include far more topics than it has, such as retrieval of optically active components, water quality monitoring, river-lake connectivity analysis, and many other issues that the authors did not capture in the search because they have in mind the hydrological processes that control and explain changes in water quantity and not quality most of the time.
Point 4:The author's scientific interest led them to write "monitoring inland water" in the sense of monitoring aquifer recharge, snowpack reduction, etc. Even so, perhaps the authors should use a more specific title regarding what is measured: water level, water surface, etc., etc., etc.
Response 2: Point 2 and point 4 raised by the reviewers are both misleading for the title of the paper.We reformulated the title according to the research topic and content of the paper, which is “Monitoring Inland Water Quantity Variations: A Comprehensive Analysis of Multi-Source Satellite Observation Technology Applications”
Point 3: When the authors use the TS= (RS OR Remote Sensing OR Satellite Altimetry OR Gravity OR GRACE) AND (River OR Fluvial OR Lake OR Glaciers OR Ice OR Snow OR Wetland OR Groundwater OR Swamp OR Marsh OR Estuary OR Bayou), they specify a specific type of Satellite Observation which is Altimetry or Gravity and that constrains the search for those type of processes that demand those types of information. When they include GRACE, they almost focus on articles dealing with GRACE applications to quantify the amount of water contained in snowpacks, underground aquifers, etc. All the other missions and sensors are compressed in two search expressions, "RS" or "Remote Sensing," which are too broad and no longer used in most of the paper as keywords.
Response 3: In this paper, satellite altimetry, remote sensing and satellite gravity are collectively referred to as multi-source satellite technology. The above satellite types contain a large number of satellites with different observation missions, but we cannot find out all the names of these satellites, so only the satellite type is set in the search terms. If there are too many search terms, the search time of the database will be greatly extended, and some documents that cannot be completely matched accurately will be missed. If there are too many search terms, the search time will be very long, and some documents that cannot be matched exactly will be missed. In order to obtain as many and accurate documents as possible in this research field, we repeatedly discussed the problem of setting search terms. The discussion part of the paper samples and judges the retrieved papers, and the results show that the accuracy rate of the literature retrieval exceeds 93%. Therefore, taking into account the efficiency and accuracy of literature retrieval, we set the above-mentioned retrieval vocabulary.

Reviewer 2 Report
It is a good contribution to the science and is a good background for future innovative methods development in the area of remote sensing application. However, its way of English writing demands editorial effort to increase its readability.
The paper has too many long sentences, misspelled and confused words, wrong or missing prepositions, and misuses of punctuation. Therefore it is suggested to have an English proofread before it is submitted as a final publishing format.
Author Response
Point 1: The paper has too many long sentences, misspelled and confused words, wrong or missing prepositions, and misuses of punctuation. Therefore it is suggested to have an English proofread before it is submitted as a final publishing format.
Response 1: In response to the opinions of the reviewers on the quality of English, we repeatedly revised the sentences of the paper and purchased the polishing service provided by MDPI.
